# Effects of Melatonin Alone or Associated with Acyclovir on the Suppressive Treatment of Recurrent Genital Herpes: A Prospective, Randomized, and Double-Blind Study

**DOI:** 10.3390/biomedicines11041088

**Published:** 2023-04-04

**Authors:** Cristiane Lima Roa, José Cipolla-Neto, Russel J. Reiter, Iara Moreno Linhares, Ana Paula Lepique, Lana Maria de Aguiar, Isadora Braga Seganfredo, Edson Santos Ferreira-Filho, Sebastião Freitas de Medeiros, Edmund Chada Baracat, José Maria Soares-Júnior

**Affiliations:** 1Discipline of Gynecology, Obstetrics and Gynecology Department, Hospital das Clínicas HCFMUSP, Faculdade de Medicina, Universidade de São Paulo, São Paulo 05403-000, SP, Brazil; 2Department of Physiology and Biophysics, Institute of Biomedical Sciences, University of São Paulo (USP), São Paulo 05508-000, SP, Brazil; 3Department of Cellular and Structural Biology, UT Health Science Center, San Antonio, TX 78229, USA; 4Biomedical Building IV—Department of Immunology, Instituto de Ciências Biomédicas—USP, Universidade de São Paulo, São Paulo 05508-000, SP, Brazil; 5Gynecology and Human Reproduction—Instituto Tropical de Medicina Reprodutiva (INTRO), Cuiabá 78043-306, MT, Brazil

**Keywords:** women, genital herpes, viruses, melatonin, acyclovir, questionnaires, quality of life, recurrence

## Abstract

Suppressive therapy of recurrent genital herpes is a challenge, and melatonin may be an alternative. Objective: To evaluate the action of melatonin, acyclovir, or the association of melatonin with acyclovir as a suppressive treatment in women with recurrent genital herpes. Design: The study was prospective, double-blind, and randomized, including 56 patients as follows: (a) The melatonin group received 180 placebo capsules in the ‘day’ container and 180 melatonin 3 mg capsules in the ‘night’ container (*n* = 19); (b) The acyclovir group received 360 capsules of 400 mg acyclovir twice a day (one capsule during the day and another during the night) (*n* = 15); (c) the melatonin group received 180 placebo capsules in the ‘day’ container and 180 melatonin 3 mg capsules in the ‘night’ container (*n* = 22). The length of treatment was six months. The follow-up after treatment was six months. Patients were evaluated before, during, and after treatment through clinical visits, laboratory tests, and the application of four questionnaires (QSF-36, Beck, Epworth, VAS, and LANNS). Results: No statistically significant difference was observed for the depression and sleepiness questionnaires. However, in the Lanns scale for pain, all groups decreased the mean and median values in time (*p* = 0.001), without differentiation among the groups (*p* = 0.188). The recurrence rates of genital herpes within 60 days after treatment were 15.8%, 33.3%, and 36.4% in the melatonin, acyclovir, and association of melatonin with acyclovir groups, respectively. Conclusion: Our data suggest that melatonin may be an option for the suppressive treatment of recurrent genital herpes.

## 1. Introduction

Genital herpes is an infectious disease caused by the herpes simplex virus (HSV), especially type II, with a recurrent character and increasing worldwide prevalence [1]. HSV remains latent in the sensory ganglia and establishes latency in neurons, evading identification by the immune system. In fact, HSV establishes latency in neurons, creating a reservoir with the potential to reactivate over the lifetime of the host [2]. Clinically, genital herpes may occur as a primary infection and as a recurrent infection. The recurrent form is characterized by mild pain, fewer lesions, and no exuberant systemic manifestations when compared with the primary infection [3]. Even so, the recurrence impairs women’s daily professional activities and relationships with their partners [4]. Therefore, clinical treatment to decrease recurrences is essential and should be performed in the long run. In this context, the first line of treatment is the antiviral acyclovir as a suppressive treatment for at least six months [5]. However, data show that viral resistance and/or therapeutic failure with this drug may occur [6]. For this reason, other drugs, such as melatonin, have been tested as a recurrent inhibitor and not as suppressive episode treatment [7].

Melatonin, produced by the pineal gland, influences several physiological processes [8] acting through G protein-coupled receptors [9], found in several central and peripheral tissues. In the immune system, the expression of melatonin receptors in the cell membrane favors the modulation of oxidative stress and the release of proinflammatory cytokines. In conjunction with cytokines, melatonin activates genital tract cell immunity, mediated by CD8 lymphocytes and natural killer cells (NK), promoting the destruction of HSV-infected cells [10]. Therefore, melatonin is an active member of the network that positively influences the innate and adaptive immune system [11,12].

Recently, a combination of melatonin, magnesium, phosphorus, and acids extracted from *Aspergillus* sp., given once a day for seven days, was compared with acyclovir 200 mg, five times a day, for five days. The study found a significant benefit with the combination containing melatonin [7]. It was also previously reported that the association of melatonin with acyclovir may present a greater benefit than the isolated use of each medication, despite their different mechanisms of action [5]. As the activity of melatonin on the recurrence of genital herpes is still unclear, the present study aimed primarily to compare the action of melatonin alone or associated with acyclovir in the suppression of recurrent genital herpes within 60 days after the end of the treatment. Secondarily, the study proposed to evaluate pain severity and quality of life of women in the three groups.

## 2. Materials and Methods

### 2.1. Design and Setting

A prospective, double-blind, randomized study was conducted at the Lower Genital Tract Pathology (PTGI) Outpatient Clinic of the Gynecology Division of the Hospital das Clínicas da Faculdade de Medicina da Universidade de São Paulo (HCFMUSP). The study protocol was registered in https://clinicaltrials.gov/ (accessed on 5 February 2019), under number NCT03831165, in February 2019. The study was approved by the Ethics Committee of HCFMUSP (CAAE: 40862215.0.0000.0068, opinion: 1.215.208). The study followed the statement of ethical principles for medical research involving human subjects from the Declaration of Helsinki.

### 2.2. Study Population

Initially, volunteers from the PTGI Outpatient Clinic, Collaborator Specialties Center, Basic Health Units, and women invited by the media (social network and newspapers) were referred for screening and selection. All participants were informed about the nature of the study and signed an informed consent form. Of these women, 90 were eligible and randomized. Because there were follow-up losses and dropouts, fifty-six women completed the study. Nineteen (33.9%) were randomized in the melatonin group; 15 (26.8%) were randomized in the acyclovir group, and 22 (39.3%) were randomized in the acyclovir with melatonin group (Figure 1).

Recruitment began on August 2018, and follow-up finished on August 2019. No interim analysis was performed, and the trial ended when all patients completed the follow-up period. Only women between 15 and 49 years with a clinical diagnosis of recurrent genital herpes were included. Pregnant or postpartum women (up to 3 months), use of oral acyclovir in the last 60 days, autoimmune diseases, immunosuppression (congenital or acquired), active liver disease, undiagnosed vulvar ulcers, severe depression (Beck inventory ≥ 30), use of antiviral drugs, immunosuppressants, antidepressants, imiquimod, allergy to melatonin or acyclovir, and women with gynecological cancer were excluded. Other eligibility criteria were the ability and willingness to provide written informed consent for the study.

### 2.3. Randomization and Masking

Simple randomization was performed through the Randomizer iOS application. Eligible women were randomized to receive acyclovir (*n* = 30), melatonin (*n* = 30), or acyclovir with melatonin (*n* = 30). The drug of each group was not known either by the patient or by the researcher until the end of the study. The allocation number of each participant was placed in a well-sealed, opaque envelope. A nurse outside the research team was responsible for distributing the medication to the participants. Names of patients and hospital were recorded in the envelopes. The envelopes were stored in an appropriate place within the institution.

### 2.4. Interventions

Melatonin and acyclovir pills were supplied by a compounding pharmacy (Essentia Pharma, São Paulo, SP, Brazil), in which acyclovir and melatonin were previously analyzed and approved according to their pharmaceutical specifications. The drugs were inserted into capsules that were not different in odor, shape, color, and size. Capsules were stored in bottles that were delivered to the participants by nurses of the Gynecology Outpatient Clinic. All women received two containers, labeled ‘day’ and ‘night’, and were instructed to take one capsule, orally, from the ‘day’ container at 10 AM and one capsule, orally, from the ‘night’ container at 10 PM for six months. The acyclovir group received 360 capsules of 400 mg acyclovir twice a day (1 capsule in the ‘day’container and another in the ‘night’ container. The melatonin group received 180 placebo capsules in the ‘day’ container and 180 melatonin 3 mg capsules in the ‘night’ container. Acyclovir with melatonin group received 180 capsules of 400 mg acyclovir in the ‘day’ container and 180 melatonin 3 mg capsules in the ‘night’ container.

### 2.5. Follow-Up and Outcomes

Eight outpatient visits were made for the study participants (Appendix A). In the first visit (screening), clinical history, possible morbidities, and variables of interest were collected as follows: age, marital status, education, profession, race, family income, religion, place of residence, sexual activity, weight, height, medical and surgical history, family history, menstrual history, sexual history, obstetric history, medications in use, health habits, and allergies. During the interview, each participant was asked about the time interval between recurrences of herpetic disease, previous treatments, and sleep quality. Physical examination of the genitalia was performed in lithotomy position, under adequate illumination, and the vulvar regions (pubic mound, labia majora and minora, vaginal introitus, and perineum) and inguinal grooves were evaluated. Vulvar lesions (vesicles, blisters, pustules, ulcers, erythema, abrasions, and scars) were identified, and cervicovaginal cytology samples were collected.

Before initiating the treatment, eligible patients received the medications and appropriate dosing guidelines. They were instructed to immediately notify the lead researcher in case of any side effects. The third, fourth, and fifth visits occurred at 30, 60, and 90 days, respectively, after the beginning of the medication, when patients were evaluated for correct use of medication, improvement or worsening of vulvar lesions (if still present in this period), adverse effects (nausea, vomiting, headache, diarrhea, drowsiness, fever, itching in the body, and spots on the body), and other possible occurrences, such as feelings of happiness or sadness, better or worse physical and psychological disposition, improvement or worsening of sexual performance, and mood change. In addition, the patients were submitted to general and gynecological physical examinations. On the sixth visit, 180 days after the beginning of the medication, possible side effects were verified and cured, and permanence or appearance of vulvar lesions were evaluated through physical examinations. Post-treatment follow-up was performed on the seventh and eighth visits, scheduled for 30 and 60 days, respectively, after the end of the medication. At this time, the results of laboratory tests, possible side effects, and disclosure of the medication used for each participant were revealed.

### 2.6. Biochemical Analysis

Blood samples were collected to perform complete blood count (evaluated in automated equipment using the panoptic staining method), blood glucose (hexokinase method), HIV serology (electrochemiluminescence method), VDRL (flocculation method), and HSV (PCR and/or chemiluminescence method, Liaison apparatus). Immunophenotyping of T lymphocytes (CD4/CD8 ratio) was performed using flow cytometry (BD FacsCalibur, BD FacsCantoII and Backman Coulter) in the Flow Cytometry Laboratory, Instituto do Câncer do Estado de São Paulo (ICESP).

### 2.7. Questionnaire Application

The investigators used patients’ responses to fill out the following questionnaires: (a) Beck’s Depression Inventory, composed of 21 categories of symptoms and attitudes that reveal cognitive and affective behaviors related to depression [13,14]; (b) Visual Analog Scale (VAS), to assess pain intensity [15]; and LANNS pain scale, to assess how innervation transmitting pain is working [16]; (c) Epworth Sleepiness Scale, to assess drowsiness (probability of napping) through eight questions [17,18]; and (d) QSF-36 (36-Item Short-Form Health Survey), consisting of a multidimensional questionnaire with 36 items, encompassing eight domains: functional capacity, physical aspects, pain, general health status, vitality, social aspects, emotional aspects, and mental health [19]. These instruments aimed to evaluate possible changes related to quality of life, quality of sleep, pain intensity, and depression in the pre- and post-treatment. Noteworthy, the Beck’s depression inventory was also used to exclude women with severe depression, which could interfere with the final results of the study.

### 2.8. Criteria for Treatment Failure and Follow-Up Failure after the Beginning of the Study

Participants who missed the outpatient scheduled visits were called by telephone or message through the WhatsApp application for new scheduling. When women did not attend the rescheduled appointment, or in the event of a change of address or telephone number without notification to the researcher or institution, she was considered as follow-up failure. Patients who became pregnant during the study, who used less than 80% of the capsules, or lost their medication, were also considered follow-up failures (Table 1).

### 2.9. Statistical Analysis

Shapiro–Wilk and Kolmogorov–Smirnov tests were used to access data distribution previously to the statistical analysis. For variables with parametric distribution, data were expressed as mean, standard deviation, median, and interquartile range (IQR) for data with nonparametric distribution. Categorical data were expressed as total numbers and proportions.

To test the hypothesis of nonmodification in the measurements among the different group interventions in time, ANOVA model was used, and the Tukey post hoc test, when applicable [20]. Wilcoxon–Mann–Whitney and Kruskal–Wallis tests were used for continuous but not normally distributed variables. Differences in the frequency distribution of categorical variables were evaluated using Chi-Square or Fisher’s exact test when appropriate.

The Relative Treatment Effects (RTEs) were calculated by ∆ treatment\∆ time. The paired *t*-test was applied to compare the recurrence of herpes between 30 and 60 days in the same group. All tests considered a bidirectional alpha of 0.05 and were performed using the R software (https://www.r-project.org (accessed on 22 September 2021), LD package for nonparametric analysis, IBM SPSS 25 (Statistical Package for the Social Sciences), and Excel 2016 ^®^ (Microsoft Office).

### 2.10. Sample Number and Power Calculation

The data on the prevalence of herpes in the the Brazilian population of women were accessed to calculate the apropriate sample number of patients to be included in this study. Based on two studies [21,22], and considering the margin level of missing patients of 20%, the calculated number of patients for this study was 28 with power of 80% and 38 with power of 90%.

## 3. Results

The median and IQR (interquartile range) age was similar among the melatonin, acyclovir, and acyclovir plus melatonin groups: 41; 32–42 years, 42; 33–48 years, and 42; 30–50 years, respectively. No significant differences were observed for age of menarche, age of first sexual intercourse, number of sexual partners, number of pregnancies, parity, abortion, or body mass index (Table 1). The sociodemographic characteristics of marital status, education, ethnicity, habits, gynecological routine, and family income did not present significant statistical differences among the groups. However, the contraceptive method was heterogeneous. While in the melatonin group all women (100%) used at least one type of contraception, in the acyclovir group, this number was six women (40%), with statistical significance (*p* = 0.010).

Figure 2 indicates that the recurrences of genital herpes at 30 days were 10.5% in the melatonin group, 13.3% in the acyclovir group, and 31.8% in the acyclovir with melatonin group. Recurrence within 60 days occurred in 15.8% of women in the melatonin group, 33.3% in the acyclovir group, and 36.4% in the acyclovir with melatonin group. Although all groups presented a similar performance in recurrence at 30 days (*p* = 0.228) and at 60 days (*p* = 0.369), a lower percentage of recurrence after 60 days was observed in the melatonin group compared to the others, although not statistically different.

Table 2 presents the evolution of some hematological parameters and fasting glycemia before and after treatment. None of the parameters showed any group versus time interaction with statistical significance. The CD4/CD8 lymphocyte ratio had little variability compared to the pretreatment moment. In the melatonin group, this ratio rose from 1.79 ± 0.71 to 1.82 ± 0.73, while in the acyclovir group, it rose from 2.10 ± 0.75 to 2.12 ± 0.88. In the acyclovir group with melatonin, although the increase was more pronounced compared to the previous ones, it ranged from 1.90 ± 0.72 to 2.02 ± 0.73, without reaching statistical significance (*p* = 0.451). Although the melatonin group recorded a higher mean evolution (2.16 ± 0.69 to 2.46 ± 0.47), no significant difference was observed among the groups.

No statistically significant difference was observed for the depression and sleepiness questionnaires. However, in the Lanns scale, for pain, all groups decreased the mean and median values in time (*p* = 0.001), without differentiation among the groups (*p* = 0.188). The VAS also presents variability in time (*p* < 0.001) and among groups (*p* = 0.002); however, none of the findings displayed any significant interaction among the groups and times (Table 3). In the quality of life evaluation before and after treatment (Appendix A), except for the domain Limitation by Emotional Aspects, none of the others presented statistical significance. In the domain that showed statistical significance, the melatonin group decreased the relative treatment effect (RTE) from 0.604 to 0.394, while in the acyclovir group, there was an increase in the RTE from 0.520 to 0.565, and in the combination group, the RTE remained unchanged.

## 4. Discussion

Systemic antiviral drugs can partially control the signs and symptoms of genital herpes when used to treat the first clinical and recurrent episodes or when used as daily suppressive therapy [23,24,25]. Nevertheless, these drugs do not eradicate the latent virus or affect the risk, frequency, or severity of recurrences after discontinuation of the drug, which is of concern to the clinician. Because HSV hides from the host’s immune system, it replicates in the nerves (sensory neurons and peripheral ganglia), resulting in a latent period with low replication, precluding HSV from being exposed to a specific cell type of immune response [26,27,28,29,30,31,32,33,34,35,36,37,38,39]. Therefore, despite the good results of antiviral drugs, the greatest challenge in the treatment of genital HSV is still recurrences.

Genital herpes infections are common, and the CDC (USA) estimates that there are 572,000 new genital herpes infections in the United States in a single year. Recurrences can occur even in those who were treated during their initial infection because treatment does not eradicate the latent virus, which can subsequently reactivate. Over time, the first recurrence may affect less than 20% of patients, but the number of subjects with a chronic recurrent form (>10 recurrences) is low (less than 0.1% of type 2 genital herpes). Those patients need chronic suppressive therapy with acyclovir for a long time, and the asymptotic period is short. Therefore, new treatments, such as melatonin, may be an option for those women [40]. Considering this possibility, the present study compared the recurrence rates of HSV after treatment with melatonin, acyclovir, or a combination of melatonin and acyclovir for six months. It was also demonstrated that the effect of melatonin is similar to that of acyclovir on the suppression of genital herpes recurrence. In addition, there was no advantage to the association between melatonin and acyclovir. In addition to that, the quality of life during the treatment period was not different among women with a different therapy. The suppressive therapy with low-dose acyclovir for a long period, commonly 400 mg twice a day, orally for six months, is well established. However, even this regimen decreases the genital transmission of HSV-2 to susceptible parties [1,34,35], and it may facilitate resistance and recurrence and is not well accepted by patients [41,42]. Furthermore, the use of acyclovir for seven days assures symptom regression in 85% of women [7]. Cases of acyclovir resistance are related to the reduction of the thymidine kinase activity [5].

In previous studies, a combination of melatonin with magnesium, phosphorus, and fatty acids extracted from Aspergillus sp for seven days has shown to improve HSV symptoms in more than 95% of patients [7], suggesting that this mixture could be superior to acyclovir. In general, no intense or severe side effects are found in women who have used melatonin. Melatonin can act on the enzyme thymidine kinase by membrane receptors or other receptors, such as two retinoid Z receptor subtypes (RZR α and β) and the three variants of other retinoid-related receptors (ROR α1, α2, and α3), which are related to the inflammatory response [43,44,45,46]. Accordingly, RZRα acts on 5-lipoxygenase, a key enzyme in the biosynthesis of leukotrienes, known to be allergic and inflammatory mediators [47,48,49] and an inhibitor of thymidine kinase expression. Aspergillus sp has shown an inhibitory effect of tobacco mosaic virus (TMV) [12]. So, it can induce cellular and humoral immunomodulation, which can influence the inflammatory process [43,44]. For this reason, the current study compared a group with melatonin alone.

Regardless of the type of treatment used in this study, there was an increase in the total number of leukocytes and lymphocytes at the end of the study. These cells are responsible for combating the virus [50]. CD4 T lymphocytes help B lymphocytes eliminate infections, being the first warning of the immune system. CD8 T lymphocytes, also known as suppressants or cytotoxic agents, decrease the activity of other types of lymphocytes and are therefore increased in cases of HIV infection [51,52]. Tryptophan precursor to melatonin plays an important role in maintaining T lymphocyte homeostasis [11], but in the current study, no significant differences among the groups of treatment, nor in relation to the baseline results of our study on genital herpes, were seen. However, it was noticed that the CD4/CD8 ratio tended to rise in the melatonin/acyclovir group, attenuating relapse crises. We did not access antigen-specific T cell responses and did not measure immune responses at shorter times during therapy. Therefore, further investigation of lymphocyte activity regarding the studied treatments will be necessary.

Regarding the questionnaires applied, the VAS and Lanns scales were used to evaluate the variable pain differences demonstrated among the groups. Because this scale may have a few limitations and be biased by depressive symptoms, the Beck scale was also applied [13,14]. In the Beck scale, the results did not show significant differences at the beginning or at the end of treatment among the participants. Regarding depressive symptoms, a few studies have shown an indirect action of melatonin, that indicated moderately lower levels in patients with anxiety disorders [53,54,55,56,57,58,59], but in women with recurrent HSV with depression and anxiety, melatonin would not be relevant, even in cases of more pronounced depression [60,61]. Therefore, increased melatonin levels can influence the results of depressive mood in women, and this can have repercussions on the immune system [62]. This may justify the significant improvement in the melatonin group in the limitations of emotional aspects of the quality-of-life questionnaire applied in this study. However, no significant differences were found in the total score of this questionnaire among the groups.

It should be considered that the recurrences themselves can cause high levels of emotional stress, emphasizing that one third of patients have more than six recurrences per year that impact on the quality of life [4]. Another aspect is that the affected women also have relationship problems with their sexual partners, which further aggravates emotional aspects [63,64,65,66,67,68,69,70,71]. All these aspects can lead to the reclusion of women who have difficulty in resuming their routine and professional and social activities. Therefore, a longer-term follow-up of these women may be necessary to assess the impact of treatments on quality of life [72]. The domain related to the limitation of emotional aspects was the one that showed an earlier improvement, especially in the melatonin group.

The SF-36 questionnaire (Short-Form Health Survey), a multidimensional instrument [63], was used to assess quality of life. In the current study, each dimension was analyzed separately, and no differences were observed among the groups, except for the domain related to depressive disorder. The use of this scale presents difficulties to access the improvement, mainly because women exposed to severe psychosocial stresses at work and in the family are the most affected by recurrent genital HSV [65]. In addition, there are chronic changes in relation to the items of the questionnaire, which are difficult to solve in the short term. A previous study showed significant improvement after five years of suppressive therapy [4].

The present study has some limitations: (a) loss of patients during the study; 16 participants did not attend the scheduled visits; 2 gave up because they were pregnant; 1 presented excessive sleepiness; 3 lost medications; 10 did not take the control tests in the correct period, and 2 did not deliver the final questionnaires. Given that all types of treatment employed are in the long term, this naturally decreases the patient’s adherence [73]; (b) some questionnaires may have subjective interpretation, which may affect the understanding of the participant; (c) possible influence of psychosocial disorders, due to the recurrent of HSV infection, may have influenced the clinical response during treatment; (d) the percentage of melatonin with recurrence during the R60 period was statistically similar to the other groups due to the low number of patients in this group with recurrence (*n* = 5). HSV disease causes significant harm to the patient‘s quality of life, while a full cure does not occur due to the viral persistence inside of the organism. Our manuscript evaluated the use of melatonin versus acyclovir both in monotherapy and in combination. It should be noted, however, that the strength of this study is that the effect of melatonin alone on genital HSV was evaluated, while another study evaluated a compound [7]. Additionally, the results show the efficacy of melatonin in reducing recurrence episodes.

In short, melatonin may be considered a therapeutic option in the treatment of recurrent genital herpes, especially in women with resistance to antiviral drugs, such as acyclovir. However, the study demonstrated that its association with acyclovir would not add any advantage. On the contrary, it could increase recurrence. Some of the factors that may have influenced the results of the present study include the inhibitory action of melatonin on thymidine kinase, which is an important enzyme for the activity of acyclovir on herpetic infection.

Finally, melatonin contributes to protecting the oral cavity from tissue damage due to the action of its receptors. The experimental evidence suggests that melatonin can be useful in treating several common diseases of the oral cavity, such as HSV disease [73]. In our trial, we reported evidence of melatonin action on HSV disease in the female lower genital tract. In fact, melatonin was similar to acyclovir in the suppression of recurrent genital herpes. Both drugs are well tolerated. When associated with acyclovir, melatonin does not seem to be more efficient than acyclovir alone. This pilot study provides results that can support larger clinical trials to validate melatonin as a new treatment regimen for patients with genital herpes, including those suffering from acyclovir-resistant strains. Further studies with larger sample sizes are needed to better understand the effects of melatonin, mainly in association with acyclovir as a suppressive treatment of recurrent genital herpes.

## 5. Conclusions

Our data suggest that melatonin may be used as an option for suppressive treatment of recurrent genital herpes.

## Figures and Tables

**Figure 1 biomedicines-11-01088-f001:**
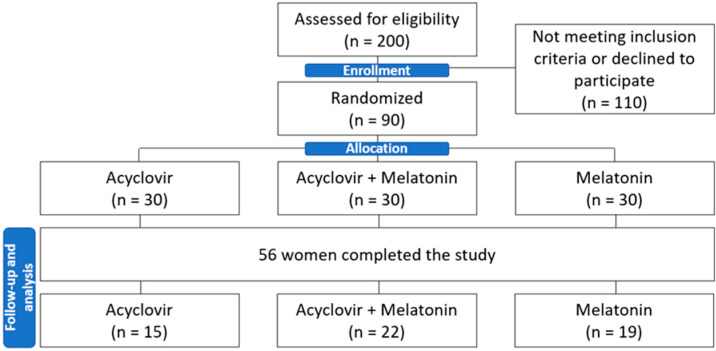
CONSORT Flow Diagram.

**Figure 2 biomedicines-11-01088-f002:**
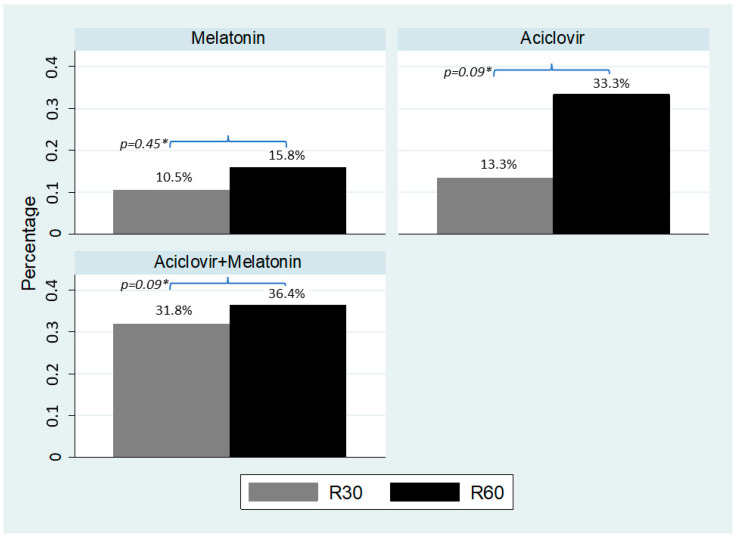
Recurrence of genital herpes at 30 and 60 days after the end of treatment among the study groups *. R30 = Recurrence at 30 days; R60 = Recurrence at 60 days; *p*-value (Fisher’s exact test): 0.228 relapses in 30-day comparison among groups; *p*-value (Fisher’s exact test): 0.369 relapses in 60-day comparison among groups. * *p*-value (Fisher’s exact test) comparison between R30 and R60 in each group.

**Table 1 biomedicines-11-01088-t001:** Sociodemographic characteristics of women evaluated according to study groups.

	Melatonin	Acyclovir	Acyclovir + Melatonin	*p*-Value *
	Median; IQR	Median; IQR	Median; IQR	
Age (years)	41; 32–42	42; 33–48	42; 30–50	0.884
BMI (kg/m^2^)	24.6; 22.0–27.2	24.5; 22.8–32,2	25.8; 22.5–28.0	0.736
Age in menarche (years)	12; 13–11	13; 12–14	13; 12–14	0.950
Age first relationship (years)	3; 17–16	18; 16–19	17; 15–20	0.934
Number of sexual partners	3; 3–1	5; 2–10	4; 3–5	0.850
Gestation	1; 1–0	1; 0–2	0; 0–2	0.943
Number of deliveries	1; 1–0	1; 0–2	0; 0–2	0.367
Number of abortions	1; 0–0	0; 0–0	0; 0–0	0.628

***** Kruskal–Wallis test; BMI = Body mass index; IQR = Interquartile range.

**Table 2 biomedicines-11-01088-t002:** Evolution of hematological parameters and fasting glycemia before and after treatment according to the research groups.

	Melatonin	Acyclovir	Acyclovir + Melatonin	*p*-Value	
	Mean ± SD	Mean ± SD	Mean ± SD	Group	Time
Fasting glucose Pre	85.68 ± 8.35	90.07 ± 6.52	92.32 ± 20.41	0.405	0.400
Fasting glucose Post	86.89 ± 9.48	90.93 ± 8.56	92.55 ± 22.5		
Hb Pre	13.1 ± 1.0	13.3 ± 0.8	12.9 ± 0.8	0.127	0.165
Hb Post	13.0 ± 0.8	13.6 ± 0.7	13.1 ± 0.7		
Ht Pre	38.95 ± 2.6	40.33 ± 2.6	38.56 ± 2.6	0.041 *	0.756
Ht Post	37.33 ± 6.4	40.81 ± 2.3	39.17 ± 1.9		
Platelets Pre	291 ± 67	253 ± 42	273 ± 57	0.155	0.428
Platelets Post	287 ± 53	253 ± 48	266 ± 57		

Pre = baseline (pretreatment) and Post = after six months of treatment. Platelets × 1.000; Hb = hemoglobin; Ht = hematocrit. * *p*-values based on the repeated measures ANOVA model followed by Tukey post hoc.

**Table 3 biomedicines-11-01088-t003:** Evolution of Beck, Lanns, VAS, and Epworth scales before and after treatment according to study groups.

Group	Time *	Mean ± SD	RTE	RTE 95%CI		*p*-Value		
				Lower	Upper	Group	Time	Group* Time
Beck for Depression								
Acyclovir + Melatonin	1	11 ± 8	0.495	0.396	0.595	0.967	0.700	0.817
Acyclovir + Melatonin	2	12 ± 10	0.496	0.390	0.602			
Melatonin	1	10 ± 7	0.497	0.395	0.600			
Melatonin	2	11 ± 7	0.491	0.378	0.605			
Acyclovir	1	12 ± 9	0.527	0.401	0.650			
Acyclovir	2	11 ± 8	0.502	0.378	0.626			
Lanns for pain								
Acyclovir + Melatonin	1	11 ± 7	0.617	0.512	0.708	0.188	0.001	0.390
Acyclovir + Melatonin	2	8 ± 6	0.500	0.407	0.592			
Melatonin	1	10 ± 7	0.602	0.490	0.700			
Melatonin	2	6 ± 5	0.430	0.331	0.537			
Acyclovir	1	6 ± 8	0.419	0.294	0.559			
Acyclovir	2	5 ± 8	0.369	0.252	0.513			
VAS for pain								
Acyclovir + Melatonin	1	6 ± 3	0.708	0.607	0.786	0.002	<0.001	0.578
Acyclovir + Melatonin	2	3 ± 3	0.502	0.380	0.624			
Melatonin	1	5 ± 3	0.564	0.472	0.649			
Melatonin	2	2 ± 3	0.282	0.208	0.380			
Acyclovir	1	7 ± 2	0.647	0.551	0.727			
Acyclovir	2	5 ± 4	0.344	0.251	0.461			
Epworth for sleepiness								
Acyclovir + Melatonin	1	10 ± 4	0.590	0.492	0.678	0.170	0.448	0.880
Acyclovir + Melatonin	2	9 ± 4	0.571	0.478	0.656			
Melatonin	1	9 ± 6	0.487	0.372	0.605			
Melatonin	2	9 ± 6	0.481	0.365	0.599			
Acyclovir	1	8 ± 4	0.405	0.298	0.526			
Acyclovir	2	8 ± 4	0.400	0.285	0.533			

*p*-values based on nonparametric repeated measures ANOVA model; RTE = Relative treatment effect. * Time: 1—baseline; 2—after treatment.

## Data Availability

Data available in a publicly accessible repository that does not issue DOIs.

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
