# Peer review of "Effects of Melatonin Alone or Associated with Acyclovir on the Suppressive Treatment of Recurrent Genital Herpes: A Prospective, Randomized, and Double-Blind Study"

_biomedicines, 2023, doi:10.3390/biomedicines11041088_

Round 1
Reviewer 1 Report
Lima Roa et al. in their manuscript "Effects of melatonin..." present the results of a small clinical study of recurrent genital herpes. Patients were treated in three groups: a) melatonin, b) acyclovir, and c) ACV + melatonin. The study was double-blinded and randomized and included 15, 22, and 19 patients, respectively. The authors found no significant differences between the groups in age, number of sexual partners, etc., but contraceptive use in the melatonin group. Surprisingly, recurrence of genital herpes 30 and 60 days after treatment was lowest in the melatonin group (15% after 60 days) compared with 33% and 36% in the ACV and ACV+mel groups, respectively.
The manuscript is well written and easy to read. The study was conducted to high standards. The only limitation of the study is the relatively small number of participants
Minor:
Please provide additional references for statement p2 lines 57-61. Reference #10 is not sufficient and not accessible.
p.1 line 43- please rewrite the sentence- “transmitter” does not sound appropriate
Author Response
Dear Reviewer,
I would like to thank you for comments. I wrote the answers below:
a) Please provide additional references for statement p2 lines 57-61. Reference #10 is not sufficient and not accessible.
We replaced the reference #10 by Carrillo-Vico A, Guerrero JM, Lardone PJ, Reiter RJ. A review of the multiple actions of melatonin on the immune system. Endocrine. 2005 Jul;27(2):189-200. doi: 10.1385/ENDO:27:2:189.
b) p.1 line 43- please rewrite the sentence- “transmitter” does not sound appropriate
We rewrote it: “HSV remains latent in the sensory ganglia and establishes latency in neurons and evading identification by the immune system. In fact, HSV establishes latency in neurons, creating a reservoir with the potential to reactivate over the lifetime of the host”.
Sincerely,
José Maria Soares Júnior
Reviewer 2 Report
The manuscript sent for review concerns an interesting topic of searching for new ways to treat recurrent genital herpes.
However, the manuscript requires corrections and additions.
Here are my comments.
1. Not all authors' affiliations are in English (l.11)
2. What does it mean in summary (frequency?) (l.27) Drug dosage is not given. In group b there appear to be 800mg of acyclovir
3. The conclusion in the summary is incomprehensible. Better conclusion in the manuscript.
4. It is hard to accept that the 15.8% of melatonin relapses are the same as the 33.3% in the acyclovir group. Despite the fact that there are no significant statistical differences. Maybe the stats need to be checked or the groups are too small.
5. What does "adhesion failure" mean (l.88)
6. "small lips? (l.137)
7. Who filled out the questionnaires -"volunteers"?? (l.165)
8. Line 204 and the entire column (Melatonin) in Table 1 is with many errors (menarche at 2 years old???
9. Table 2, apart from possibly the level of leukocytes, does not add anything, it is unnecessary.
10. In table 3 What does Time “1” or “2” mean?
The manuscript may be published after corrections and additions
Author Response
Dear Reviewer,
Thanks for your comments and I wrote the response of questions below:
a) Not all authors' affiliations are in English (l.11)
We doubt checked it. Only the Medical School University of São Paulo standardized the name of institutional in Portuguese for international citation ( Hospital das Clínicas, Faculdade de Medicina da Universidade de São Paulo). I replace the "Disciplina de Ginecologia" and "Departamento de Obstetrícia e Ginecologia" by the english version.
b) What does it mean in summary (frequency?) (l.27) Drug dosage is not given. In group b there appear to be 800mg of acyclovir.
We corrected them.
c) The conclusion in the summary is incomprehensible. Better conclusion in the manuscript.
We replaced it for the conclusion in the manuscript.
d) It is hard to accept that the 15.8% of melatonin relapses are the same as the 33.3% in the acyclovir group. Despite the fact that there are no significant statistical differences. Maybe the stats need to be checked or the groups are too small.
We agree and double checked it. We included it on the discussion as limitation of study.
e) What does "adhesion failure" mean (l.88)
I replace it for dropouts.
f)"small lips? (l.137)
Large and small lips were replaced by Labia majora and minora.
g) Who filled out the questionnaires -"volunteers"?? (l.165)
It was mistype. We replaced volunteers by investigators.
h) Line 204 and the entire column (Melatonin) in Table 1 is with many errors (menarche at 2 years old???
We corrected the. We replaced “2” by the “12”
i) Table 2, apart from possibly the level of leukocytes, does not add anything, it is unnecessary.
We removed it from table 2
j) In table 3 What does Time “1” or “2” mean?
We included in the table: Time 1 = baseline; Time 2 = after treatment.
Sincerely,
José Maria Soares Júnior
Correspondent author
Reviewer 3 Report
- The use of Aspergillus sp alone can be excluded from the introduction. This information does not looks related to the article`s topic.
- The article is devoted to important problem of global healthcare as herpes infection treating methods. This disease causes significant harm to the patient`s quality of life while a full cure does not occur due to the viral persistence inside of the organism. The article presents the results of a clinical study conducted in Brazil on the use of melatonin versus acyclovir both in monotherapy and in combination.
- Data about the anti-HSV effect of melatonin are extremely limite. Some articles describe the effects of melatonin, which can potentially have a positive effect on patients with oral herpes (DOI: 10.1016/j.archoralbio.2011.03.004 ) Thus, the study really addresses a specific gap in the field.
- The present study demonstrates ways to find an alternative to standard anti-HSV therapy. The results obtained will be able to form a new treatment regimen for patients with genital herpes, including those suffering from acyclovir-resistant strains.
- The most significant improvement that could be recommended to the authors is to increase the number of patients to improve the statistical significance of the results obtained. It would also be possible to focus on the data obtained with the combined use of acyclovir and melatonin and make assumptions about the reasons for the increase of recurrences number in the corresponding group.
- The conclusions are consistent with the results obtained.
- The references appropriate
Author Response
Dear Reviewer,
Thanks for your comments that we answered below:
a) The use of Aspergillus sp alone can be excluded from the introduction. This information does not looks related to the article`s topic.
We removed it.
b) The article is devoted to important problem of global healthcare as herpes infection treating methods. This disease causes significant harm to the patient`s quality of life while a full cure does not occur due to the viral persistence inside of the organism. The article presents the results of a clinical study conducted in Brazil on the use of melatonin versus acyclovir both in monotherapy and in combination.
We included it in the discussion for reinforcing the strengths of study.
c) Data about the anti-HSV effect of melatonin are extremely limited. Some articles describe the effects of melatonin, which can potentially have a positive effect on patients with oral herpes (DOI: 10.1016/j.archoralbio.2011.03.004 ) Thus, the study really addresses a specific gap in the field.
We included it on discussion.
d) The present study demonstrates ways to find an alternative to standard anti-HSV therapy. The results obtained will be able to form a new treatment regimen for patients with genital herpes, including those suffering from acyclovir-resistant strains.
We included this statement on the discussion.
e) The most significant improvement that could be recommended to the authors is to increase the number of patients to improve the statistical significance of the results obtained. It would also be possible to focus on the data obtained with the combined use of acyclovir and melatonin and make assumptions about the reasons for the increase of recurrences number in the corresponding group.
We included it in the limitations and our study was pilot study.
Sincerely,
José Maria Soares Júnior
Correspondent author